# Higher Dietary Intake of Advanced Glycation End Products Is Associated with Faster Cognitive Decline in Community-Dwelling Older Adults

**DOI:** 10.3390/nu14071468

**Published:** 2022-03-31

**Authors:** Michal Schnaider Beeri, Roni Lotan, Jaime Uribarri, Sue Leurgans, David A. Bennett, Aron S. Buchman

**Affiliations:** 1Department of Psychiatry, The Icahn School of Medicine at Mount Sinai, New York, NY 10029, USA; 2The Joseph Sagol Neuroscience Center, Sheba Medical Center, Tel Hashomer 52621, Israel; lotanronnie@gmail.com; 3Department of Medicine, The Icahn School of Medicine at Mount Sinai, New York, NY 10029, USA; jaime.uribarri@mssm.edu; 4Rush Alzheimer’s Disease Center, Rush University Medical Center, Chicago, IL 60612, USA; sue_e_leurgans@rush.edu (S.L.); david_a_bennett@rush.edu (D.A.B.); aron_s_buchman@rush.edu (A.S.B.); 5Department of Neurological Sciences, Rush University Medical Center, Chicago, IL 60612, USA

**Keywords:** advanced glycation end-products, diet, cognitive decline

## Abstract

Objective: Dietary-derived advanced glycation end products (AGEs) vary for different food types and the methods employed during their preparation may contribute to diverse chronic health conditions. The goal of this study was to investigate the associations of dietary AGEs (dAGEs) with cognitive decline in older adults. Methods: Non-demented older adults (*n* = 684) underwent annual testing with 19 cognitive tests summarized as a global cognitive score based on five cognitive domains. We modified a previously validated food frequency questionnaire designed to assess dAGE. The modified questionnaire assessed portion size and frequency of consumption of six food groups (meat, poultry, fish, cheese, spreads, and processed foods), as well as the method of their preparation (e.g., grilling, boiling). dAGE was the sum of the scores of the six food groups. Linear mixed-effect models were used to examine the association of baseline dAGE with cognitive decline. All models controlled for age, sex, education, race, and body mass index (BMI). Results: Average follow-up was 3.0 years. Higher baseline dAGEs was associated with a faster rate of global cognitive decline (Estimate = −0.003 (standard error = 0.001, *p*-value = 0.015). This association was driven by declines in episodic memory (−0.004 (0.002, 0.013)) and perceptual speed (−0.003 (0.001, 0.049)) but not by semantic memory, working memory, and visuospatial domains. These associations were not attenuated by controlling for cardiovascular risk factors and diseases, including diabetes. Levels of dAGE of the specific food groups were not associated with cognitive decline. Conclusions: Higher levels of dietary AGE levels in older adults are associated with faster cognitive decline. These data lend further support for the importance of diet and that its modification may slow or prevent late-life cognitive impairment. Further clinical studies will be needed and the molecular mechanisms underlying these associations will need to be identified.

## 1. Introduction

Alzheimer’s disease (AD) dementia is the most common cause of late-life dementia in older adults. Since there are no robust medications to treat AD dementia, intense efforts are focused on identifying potentially modifiable lifestyle factors that can be employed to maintain cognition and brain health in older adults. For example, much research has focused on late-life behaviors like cognitive, physical, and social activities which may maintain or slow cognitive decline. However, affecting changes in longstanding lifestyles in older adults is particularly challenging. Other lifestyle factors may be more amenable to modification in older adults. There has been increasing focus on dietary consumption and recent clinical studies suggest that specific diets, such as the Mediterranean, Mediterranean-DASH Intervention for Neurodegenerative Delay (MIND), and Dietary Approaches to Stop Hypertension (DASH) diets, may be associated with better cognition in older adults [1,2,3].

However, not only the types of foods but also how they are processed may contribute to cognition in older adults. There is increasing evidence that higher levels of advanced glycation end products (AGEs) in the body may contribute to diverse chronic conditions [4]. AGEs content of food consumed by older adults varies with the food type and methods employed for its preparation. Thus, the level of AGEs in daily food consumption may contribute to the level and rate of cognitive decline in older adults. AGEs are formed endogenously in the course of normal cellular metabolism and their levels tend to increase with age [5]. Exogenous sources such as pre-formed AGEs from food may also contribute significantly to the body’s pool of AGEs [6,7]. AGEs exist in most foods, but they are particularly abundant in animal-derived foods, such as meat, poultry, and fish, which are also rich in fats. Content of AGEs in these foods increases significantly when cooking at high temperatures using a dry cooking technique (e.g., grilling, broiling, roasting, searing), but less when using lower temperatures and a cooking technique with more moisture (e.g., boiling, poaching). The content of AGEs is much lower in plant-derived foods [4,8,9]. Therefore, modifying the type of food consumed and the techniques employed for its preparation can affect levels of exogenous dietary AGEs. There are few studies examining the association of dietary advanced glycation end products (dAGEs) consumption with cognitive decline.

The association of intake of dietary AGEs and cognition has been studied using very small samples (*n* = 48) [10] or by assessing dietary AGEs indirectly using general food frequency questionnaires [11] which were not designed to measure dietary AGEs intake. To address this knowledge gap, we modified a previously validated dietary AGEs consumption questionnaire so that it could be deployed in the Rush Memory and Aging (MAP) project, an ongoing community-based cohort study of aging and dementia in older adults. We then examined the association of dAGEs consumption with cognitive decline in 684 participants followed for an average of three years [12].

## 2. Methods

### 2.1. Participants

All participants were from MAP, a longitudinal cohort study of chronic aging conditions that started in 1997 [13]. Participant recruitment from over 40 facilities in metropolitan Chicago included subsidized senior housing facilities, retirement communities, retirement homes, social service agencies, and church groups. Participants provided written consent and signed a repository consent to all their data being repurposed. MAP participants were included if they were older than 65 years of age, had no known dementia, agreed to annual assessments and brain, spinal cord, selected muscles, and nerve autopsy at the time of death. To reduce the burden and enhance follow-up, assessments were performed at the participant’s residence. The study was approved by the Institutional Review Board of the Rush University Medical Center (MAP ORA# L86121802). The dAGE data collection began in 2017. The baseline for this study was defined as the first visit with a valid dAGE measure. Of 943 MAP participants with any dietary assessment, we successively excluded 43 participants with no valid dAGE, 72 who had clinical dementia at baseline, 34 with missing data in the primary analytic model for cognitive decline, and 110 without follow-up testing. The to arrive at an analytic sample of 684 participants of whom 202 had one follow up, 352 had two follow ups, and 130 had three follow ups. 

### 2.2. Cognition and Cognitive Status Assessment

At the uniform clinical testing performed annually, a trained research assistant administered 21 cognitive tests scored by a computer, of which 19 were used to construct a summary global cognitive score and five cognitive domain scores: episodic memory, semantic memory, working memory, perceptual speed, and visuospatial ability, as described in prior publications [13,14]. The Mini-Mental State Examination [15] was used for descriptive purposes and for diagnostic classification. Cognitive test scores were first converted to z-scores using the baseline mean and standard deviation for the entire cohort. After reversing if necessary to make positive z’s represent good cognition, the 19 z-scores were averaged to obtain the global cognition score. The skewness and kurtosis of the cognitive composite at the first visit are −0.402 and −0.047 suggesting a normal distribution. Scores for each of the five cognitive domains were calculated similarly. 

Cognitive impairment was assigned by a neuropsychologist who reviewed the cognitive test results. Finally, using the clinically available data and all cognitive testing data, a clinician diagnosed dementia and Alzheimer’s dementia based on the recommendations of the joint working group of the National Institute of Neurologic and Communicative Disorders and Stroke and the Alzheimer’s Disease and Related Disorders Association [16]. A diagnosis of Mild Cognitive Impairment (MCI) was rendered if the participant was judged cognitively impaired by the neuropsychologist but did not meet dementia criteria by the clinician [17]. Each annual diagnosis was made blinded to data from prior assessments (see additional details on procedures to assess cognitive impairment in the Appendix A).

### 2.3. Assessment of Dietary AGEs

To minimize participant burden, we modified a previously validated dAGE questionnaire [18]. We retained a single question for each of the 6 food categories assessed. For example, in the detailed questionnaire, each type of meat received its own score which accounted for the (a) size of the portion consumed, (b) how the food was prepared and (c) frequency of its consumption (e.g., a participant who reported a one-time consumption of 5 ounces of grilled steak and a two-time consumption of 4 ounces of meatballs in sauce). In the modified version of the questionnaire, we asked a single question about all types of meat that might be consumed and used the most common meat to calculate the score for the meat category.

The modification yielded 24 questions that assessed six food categories (meat, fish, poultry, cheese, spreads, and processed foods) to obtain: (a) the consumption frequency during the past week, (b) the estimated portion sizes, and (c) how the food was prepared (roasting, boiling, frying, broiling/grilling, or canned). Each food item was assigned an advanced glycation end product (AGE) value based on a database of ~560 foods that list AGE values expressed as AGE Equivalents (Eq/day) (1 AGE equivalent = 1000 kilounits) [9], which has been developed by our group. The AGE value for each food was multiplied by the portion size and also using a preassigned value for the different ways the food could be prepared. The average AGE scores for the six categories were used as the daily dAGE score. The modified AGE dietary questionnaire employed in this study and its scoring methods is included in the Appendix A.

To minimize errors and missing data, the modified dAGE questionnaire was administered by research assistants who were trained by a nutritionist, rather than participants filling it out themselves. Since this questionnaire relies on self-report, only responses from participants without clinical dementia were included in the analyses. The questionnaire was administered to the participant alone if he/she cooked their own food, or together with the person (e.g., spouse) who prepares the participant’s meals.

We compared the dAGE score obtained with the modified dAGE questionnaire with the more detailed original dAGE questionnaire. To obtain a modified dAGE score, we applied the scoring method of the simplified questionnaire on the detailed questionnaire data from 75 older adults. The Pearson correlation between total daily dAGE in the full and modified questionnaires was 0.93 (*p*-value < 0.001). The individual food categories were also strongly correlated ranging from 0.90–0.98.

### 2.4. Other Covariates

Demographic covariates were age, sex, years of education, and race. We also covaried for body mass index (BMI) (calculated from weight/height [2]), a commonly used measure of nutritional status in older adults [19]. Our models included the number of three self-reported vascular risk factors (hypertension, type 2 diabetes, and smoking) and four vascular diseases (heart attack, congestive heart failure, claudication, and stroke) as described in prior publications [20]. Hypertension was also indicated by current use of antihypertensive medication, or measured systolic/diastolic blood pressure ≥160/90 mm Hg. The current medication also indicated diabetes and heart attack (cardiac glycosides, e.g., digoxin, Lanoxin, etc.). Smoking was defined as never/ever. History of claudication was indicated by self-reported pain in calves while walking. History of stroke was also indicated by cognitive testing, interviews with participants, and neurological examination (when available).

### 2.5. Statistical Analyses

The associations of dietary AGEs with age, education, and BMI were tested with Pearson correlations. Student t-test compared measures between men and women and race. Linear mixed models [21] were employed to examine whether dAGEs were associated with baseline and cognitive decline, the clinical hallmark of AD dementia. The core model consisted of terms for baseline dAGEs score, time measured in years since first dAGE assessment (representing an annual rate of change in global cognition score), and their interaction, and terms for age, sex, education, race, and BMI, and their interactions with time. Secondary analyses were adjusted also for cardiovascular risk factors and diseases. As global cognition is constructed from five cognitive domains (episodic memory, semantic memory, working memory, perceptual speed, and visuospatial domains), we used similar models to examine if the association of dietary AGEs varied with the rate of decline of these different cognitive domains. Models were examined graphically and analytically, and assumptions were judged to be adequately met. Programming was done in SAS v. 9.4 (SAS Institute, Inc., Cary, NC, USA).

## 3. Results

### 3.1. Description of the Analytic Cohort

There were 684 non-demented older adults included in these analyses with an average follow-up of about 3 years (2.90 years (standard deviation (SD) = 0.69 years). The average participant was 82.5 years old and had 16 years of education, 3/4 of the sample were female, and the sample was on average overweight (BMI = 27.6). Additional clinical characteristics for the analytic cohort are summarized in Table 1. The dAGEs score ranged from 0.43–48.6 (Mean = 16.0; SD = 8.6) with a lower score indicating lower dAGEs consumption. The distribution of the dAGE scores was suitable for regression models (skewedness = 0.84, kurtosis = 0.91). The average levels for each of the six food categories are provided in Table 1. The correlations among the different dAGE categories was small (Appendix A; *r* < 0.19). A lower dAGEs score was related to older age (r = −0.12; *p* = 0.001) and lower BMI (*r* = 0.14; *p* < 0.001). dAGEs was lower in women (Women: mean = 15.31; SD = 7.50) versus (Men: mean = 18.33; SD = 9.29) [t(682) = −3.72, *p* < 0.001). dAGEs were not associated with number of vascular disease or risk factors, years of education, or race. dAGEs was similar in individuals with and without diabetes (t(682) = 0.28, *p* = 0.78).

### 3.2. Associations of Dietary AGEs with Baseline Cognition and with Cognitive Decline

We employed a mixed regression model adjusting for age, sex, education, race, and BMI, to assess the association of dAGE with cognitive decline. In this model, on average, cognition declined by 0.07 units per year (Table 2, Term of “Time”). On average, the level of global cognition was lower in individuals with MCI as compared to those with no cognitive impairment. Baseline levels of dAGE intake were not associated with baseline global cognition (Table 2, Term of “daily dAGE”). However, when examining the association of the dAGEs score with the annual rate of cognitive decline, we found that a higher baseline dAGEs score was associated with a faster rate of cognitive decline (Table 2, Term of “Time X dietary AGEs”). Figure 1 illustrates the differences in the rate of average cognitive decline for two types of typical participants, i.e., female, 82.5 years old with 15.9 years of education with high and low dAGE. Those with a high dAGE (90th percentile) show a 50% faster rate of cognitive decline compared with low dAGEs (10th percentile). The associations were unchanged when including terms for vascular risk factors and vascular diseases (see Appendix A).

To examine whether a specific food category was driving the association of dAGE with cognitive decline, we repeated the analyses and examined the associations of dAGE intake of each food category with global cognitive decline. None of the specific food categories was significantly associated with cognitive decline (see Appendix A for a summary of these results).

Our measure of global cognition summarized five different cognitive domains. We examined if the association of baseline dietary AGEs intake showed differential associations with the rate of decline of the five cognitive domains. dAGE intake was significantly associated with a faster decline in episodic memory and perceptual speed suggesting that these cognitive domains drove the findings with global cognition (Table 3). dAGE intake was not associated with the rate of decline in working memory, semantic memory, and visuospatial domains. Further adjustment for vascular risk factors and diseases did not alter these results (see Appendix A).

## 4. Discussion

This study of nearly 700 well-characterized community-dwelling older adults without dementia at baseline provides evidence that higher baseline dietary AGEs intake is associated with a faster rate of cognitive decline. Adjustment for sociodemographic variables, BMI, and cardiovascular risk factors and diseases did not attenuate this relationship. In the current analysis total, dAGE was associated with cognitive decline, even though no associations were identified with specific categories of dAGEs. This suggests that there is not just one specific category that drives the association with cognitive decline, but rather, that the total daily AGEs may be most important. This result may also suggest that larger samples will be necessary to detect associations with the specific dAGE categories. Further analyses of five different cognitive domains suggest that higher baseline dAGEs were associated with the rate of declining episodic memory and perceptual speed but not with semantic memory, working memory, and visuospatial domains. Thus, dietary AGEs may affect some cognitive domains more than others. Our results provide support for the growing recognition of the potential role of modifiable dietary factors like AGEs in the brain health of older adults. Further studies are needed to identify mechanisms underlying these associations to inform on the design of clinical trials to translate these findings into clinical practice [4].

In recent years there is a growing interest in the effects of nutrition, a modifiable risk factor, on cognitive decline in old age. Specifically, the Mediterranean Diet [2], DASH diet [3], and the more recently established MIND diet [1] have shown promising associations with cognition. These diets focus on the intake of specific types of foods that may have either a potential neuroprotective (such as vegetables [22] and fish [23]) or neurotoxic effects (such as foods high in saturated fats [24]). Prior work has focused on the type and quantity of food types with a paucity of data about how the food was prepared. Circulating levels of AGEs, which contribute to diverse adverse health outcomes [4], are affected by the dietary consumption of AGEs. Cooking a wide range of food types, particularly animal-derived foods, at high temperatures, such as frying or grilling, generate higher levels of AGEs as compared to cooking non-meat products or foods cooked with methods that use lower temperature such as boiling or steaming. The dAGE questionnaire developed for this study employed several of the basic features of conventional food frequency questionnaires used to assess the Mediterranean, DASH, and MIND diets including the quantity and frequency of food type consumption used. However, the dAGE questionnaire includes questions specifically addressing the cooking methods. The modifications in the dAGE of a prior longer version were made to facilitate the deployment of the dAGE questionnaire in an ongoing cohort study of large numbers of community-dwelling older adults. The modified dAGE was strongly correlated with the original validated questionnaire which has been applied in prior smaller studies [7,9]. The current study extends these prior studies showing in a large sample of older adults that higher estimated levels of dAGE were associated with faster cognitive decline.

The biology underlying the association between higher dietary levels of AGEs and cognitive decline is unclear but could result from either the adverse systemic consequences of elevated circulating AGEs or through specific effects on central nervous system (CNS) structures. Through crosslinking of proteins or activation of the receptor for AGEs (RAGE), the systemic burden of AGEs increases oxidative stress and inflammation [25] which play a major role in chronic diseases such as atherosclerosis [26] and cardiovascular risk factors [27], as well as in metabolic derangements, including diabetes [28], all of which have been associated with cognitive decline. Several cross-sectional studies have suggested that elevated serum AGEs are associated with poorer cognition in older adults [29,30,31]. Two longitudinal studies have examined the association of systemic levels of AGEs with cognitive decline, one focused on methylglyoxal [32] and another on pentosidine [33], and found a faster cognitive decline in older adults. Although serum AGE levels are consistently higher in diabetes, the results of these studies were similar in individuals with and without diabetes suggesting that the pathways via which AGEs may affect the brain do not necessarily require systemic metabolic derangement. Our results further support this possibility, as adjusting for cardiovascular risk factors and diseases, including diabetes, as well as for BMI, did not alter the associations of dAGE with cognitive decline. However, numerous factors related to metabolic conditions such as diabetes (e.g., medications, co-morbidities) may mask their role in the association of AGEs with cognition. Interestingly, recently, numerous studies have shown that high AGE levels are associated with poor muscle function [34]. While muscle function has been consistently associated with cognitive decline and incident AD [35,36] the basis for this association is unknown. Since muscle plays a crucial role in systemic metabolism, further work will be needed to determine to what extent AGEs may degrade motor function via negative effects within CNS structures underlying CNS motor control systems and/or via peripheral muscle. Several neuropathological mechanisms may underlie the association of dAGE with cognitive decline. We have recently shown that AGE levels in the serum are correlated with brain levels of AGEs suggesting that circulating AGEs cross readily the blood-brain barrier (BBB), which in turn, has been shown to be impaired in the presence of AGEs [37,38] suggesting a vicious cycle. In addition, mice fed a high AGE diet had significantly higher hippocampal levels of AGEs and β-amyloid [6], and greater mitochondrial dysfunction [39] compared to controls. Human brain imaging studies suggest that AGEs may accelerate regional cortical atrophy [40].

Furthermore, AGEs may also affect cognition via accelerated cerebrovascular disease, as elevated systemic AGEs are associated with carotid atherosclerosis [41]. However, in our study, adjusting for cardiovascular risk factors and diseases, including stroke, did not alter the results suggesting other pathological mechanisms. In fact, no study has investigated the molecular mechanisms underlying the associations identified in the current study between dAGE and cognitive decline. Further work is needed to elucidate these crucial knowledge gaps to facilitate AGE targeted interventions that may contribute to brain health and maintenance of good cognition in older adults.

Our study has limitations. The data derives from a select cohort that agreed to autopsy at death and may differ in important ways from older persons in the general population such as education, socio-economic status, and most pertinently to this study, lifestyle and cooking methods. It will be important to replicate these findings in more diverse cohorts. The adults in this study were very old and our results may not reflect the associations that might be observed in midlife or younger older adults. Endogenous AGEs increase with age [5] and it is difficult to separate the contribution of dAGEs to the endogenous AGEs pool. The analyses in this study adjusted for age to account for its potential confounding effect. Yet, further, studies will be needed which validate these findings with concurrent serum measures of AGEs. As in most large-scale studies investigating the association of diet with health outcomes, dietary data is self-reported rather than directly measured. Hence, under-reporting may have occurred, possibly suggesting even stronger associations with more objective measurements of dAGEs. Dietary AGEs have been consistently associated with serum levels of AGEs [6,7] so their further validation against serum AGE levels is needed. It is possible that reported levels of dAGEs may include other dietary components contributing to cognitive decline, underscoring the importance of replicating our findings with more objective measures of dAGEs. The follow up of this study averaged three years. As follow up continues, we will be able to examine associations of dAGEs intake with incident MCI and AD. Finally, our results remained robust after adjusting for BMI, a commonly used measure of nutritional status, and with cardiovascular risk factors and diseases such as diabetes. But, other factors such as cancer may have affected dietary intake, including AGEs. Confidence in the findings from this study is enhanced by several factors. A large number of male and female participants were examined annually for an average of three years with structured validated clinical measures of cognitive function and a food frequency questionnaire specifically tailored to measure dietary AGEs.

To conclude, we validated an easy-to-administer questionnaire to assess daily AGEs consumption and found that higher levels of intake of dAGEs are associated with a faster rate of cognitive decline controlling for demographic and cardiovascular risk factors. Since the dietary intake of AGEs is modifiable, our results based on observational data highlight the need for further clinical trials to provide evidence that modifying dietary AGEs may slow cognitive decline in older adults.

## Figures and Tables

**Figure 1 nutrients-14-01468-f001:**
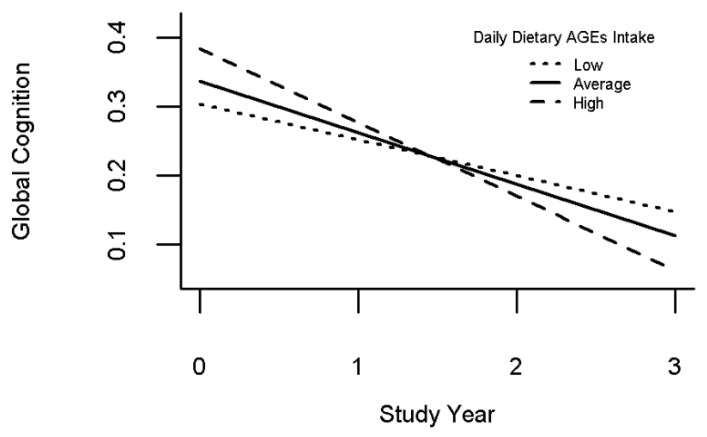
Higher dietary intake of advanced glycation end-products (AGEs) is associated with faster cognitive decline. Rate of cognitive decline in participants with high (90th percentile), average (50th percentile), and low (10th percentile) baseline AGEs.

**Table 1 nutrients-14-01468-t001:** Clinical Characteristics at Study Baseline (*n* = 684).

Clinical Characteristics at Study Baseline (*n* = 684)	
Age, years	82.46 (7.24)
Women, %	77.05
Education, years	15.93 (3.02)
Race, % White	93.9%
BMI, kg/m^2^	27.61 (5.41)
Vascular risk factors, %	
Hypertension Diabetes Smoking (ever)	57.31 16.81 43.71
Vascular disease, %	
Claudication Stroke Heart attack Congestive heart failure	13.01 9.36 5.70 5.99
Dietary AGEs	
Daily total Meat Poultry Fish Cheese Spreads Processed foods	16.00 (8.04) 27.44 (25.77) 27.00 (22.04) 12.91 (15.37) 12.42 (11.17) 11.30 (7.85) 21.02 (23.56)

BMI, body mass index; AGEs, advanced glycation end products.

**Table 2 nutrients-14-01468-t002:** Dietary AGEs intake is Associated with Decline in Global Cognition.

Dietary AGEs Intake is Associated with Decline in Global Cognition	
Model Term	Estimate (S.E, *p*-Value)
Time	−0.077 (0.011, <0.001)
Age	−0.027 (0.003, <0.001)
Sex	−0.140 (0.045, 0.002)
Years of Education	0.040 (0.006, <0.001)
Race	−0.472 (0.087, <0.001)
BMI	0.004 (0.003, 0.254)
Daily dAGE	0.004 (0.002, 0.100)
Time*daily dAGE	−0.003 (0.001, 0.015)

This model shows the results for the model terms of “Time” i.e., the annual rate of change in global cognition and the interaction of baseline dietary AGEs score with “Time” (Time*daily dietary AGE (dAGE)) from a single linear mixed-effect model which also included additional terms for age, sex, education, race, BMI, and their interaction with time (not shown). S.E, standard error.

**Table 3 nutrients-14-01468-t003:** Dietary AGEs intake and declining cognitive domains.

Dietary AGEs Intake and Declining Cognitive Domains		
Cognitive Ability Outcome	Model Term	Estimate (S.E, *p*-Value)
Global Cognition	Time	−0.078 (0.011, <0.001)
	Time × dietary AGEs intake	−0.003 (0.001, 0.015)
Semantic Memory	Time	−0.072 (0.013, <0.001)
	Time × dietary AGEs intake	<0.001 (0.001, 0.923)
Episodic Memory	Time	−0.080 (0.015, <0.001)
	Time × dietary AGEs intake	−0.004 (0.002, 0.015)
Working Memory	Time	0.021 (0.016, 0.193)
	Time × dietary AGEs intake	−0.001 (0.002, 0.380)
Perceptual Speed	Time	−0.117 (0.013, <0.001)
	Time × dietary AGEs intake	−0.003 (0.001, 0.049)
Visuospatial Abilities	Time	−0.041 (0.018, 0.027)
	Time × dietary AGEs intake	−0.001 (0.002, 0.449)

## Data Availability

All data analyzed in this study are available through the Rush Alzheimer’s Disease Center Research Resource Sharing Hub, https://www.radc.rush.edu (14 February 2022), which has descriptions of the studies and available data. Any qualified investigator can create an account and submit requests for deidentified data.

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
