# Peer review of "Higher Dietary Intake of Advanced Glycation End Products Is Associated with Faster Cognitive Decline in Community-Dwelling Older Adults"

_nutrients, 2022, doi:10.3390/nu14071468_

Round 1

Reviewer 1 Report

This manuscript entitled “Higher dietary intake of advanced glycation end products is associated with faster cognitive decline in community dwelling older adults” has been reviewed sincerely. I felt that the work was of interest and well-performed. However, I think many confounding factors should be considered in the study design. Authors should perform a multivariate analysis among the significant factors with 2 variable analysis or adjust the confounding factors statistically so that they can extract factors affecting cognitive effects independently.

Below I would like to list other some of my suggestions;

  1. It is well known that Advanced Glycation End Products (AGEs) is associated with age. I think this may be endogenous. It is difficult to distinguish between endogenous and dietary intake (exogenous) in the increase of AGEs clearly. The physiological increase is an important concern in this study. Suitable additional chemical examinations or comparative study may be needed.
  2. It is important to examine whether there were significant associations between severity of cognitive dysfunction and dietary AGEs intake.
  3. Number of subjects including drop out should be described in Methods section, but not Results section. How about exclusion criteria?

Reviewer 2 Report

Explain in the abstract what dietary AGEs are. You need to say that the preparation of food is affecting the AGE content.

line 44: "specific diets", such as MIND, meditarrean, DASH 
lines 50-55: you start talking about "animal derived foods". I am thinking milk and cheese. Then you talk about grilling. I am thinking about meat. Then you talk about food preparation in general. I am thinking about vegetables. You need to be very clear what you are focusing on. If the whole point is about preparation of meat and fish then that should be mentioned. If you include grilled cheese or halloumi, then you need to say this. If grilled broccoli does not fall under this then change the "types of foods" in line 55 to the food category that is appropriate, which at the moment I think is high-fat meats.
line 58: give typical sample sizes and give references
line 59: "were not designed to measure dietary AGEs intake" give references
line 61: add "Project" to Memory and Aging

Methods section
Was there a minimum age limit? For example, people aged 60+ were included.
Z-score transformation assumes that the data was normally distributed. Please, provide Kolgomorov-Smirnov statistics to demonstrate this, as performance data are rarely normally distributed.
line 98: a neuropsychologist assigned cognitive impairment based on the cognitive test results. What was the criteria? I get the impression that no threshold was used and that only one person did this (no available inter-rater reliability).
line 99: "select clinical data" Is this the unknown test in line 91?
line 119: ssigned
line 124: seven categories ?
line 126: There is no electronic supplemental material. Please put the information in an Appendix.

Results
line 187: mention how Time was coded.
line 192: There is no Figure 1
lines 198-202: What does this mean? None of the specific categories is associated with cognitive decline, but the total is? Please provide the covariance matrix of the categories.
lines 197, 201, 210: as there is no electronic supplement, I am unable to comment on what might be in there. For the revision I advice to add the supplement to the paper.

Discussion
lines 217-219: the lack of specific effects does not suggest that all categories contribute. If that was the case that all categories should have shown a specfic effect. There are several other possibilities that can explain this pattern. Ideally, the authors would have made their data available other researchers to analyse, but there is no mention of data access
line 235: "Cooking a wide range of food types at high temperatures..." Again, non-meat products come to mind. 
line 237: you mention boiling, but obviously you do not boil a steak. Therefore you could be assuming non-meat products, such as vegetables, but then you would not boil nutrients out of it, but instead steam them. Hence "steaming" should be included. But in general, you need to be clear about to what food you are referring.
line 249: typo in "results"

Round 2

Reviewer 1 Report

None